

# Transport dynamics in a complex coastal archipelago

Elina Miettunen[1], Laura Tuomi[2], Antti Westerlund[2], Hedi Kanarik[2], and Kai Myrberg[1]

[1] Finnish Environment Institute, Latokartanonkaari 11, FI-00790 Helsinki, Finland
[2] Finnish Meteorological Institute, P.O.Box 503 , FI-00101 Helsinki, Finland

**Correspondence:** Elina Miettunen (elina.miettunen@syke.fi)

**Abstract.** The Archipelago Sea (in the Baltic Sea) is characterised by the complex geometry of thousands of islands and steep gradients of the bottom topography. Together with the much deeper Åland Sea, the Archipelago Sea acts as pathway to the water exchange between the neighbouring basins, Baltic proper and Bothnian Sea. We studied circulation and water transports in the Archipelago Sea using a new high-resolution NEMO configuration that covers the Åland Sea–Archipelago Sea area with horizontal resolution of around 500 m. The results show that currents in the area are steered by the geometry of the islands and straits and the bottom topography. Currents are strongest and strongly aligned in the narrow channels in the northern part of the area, the directions alternating between south and north. In more open areas, the currents are weaker with wider directional distribution. During our study period of 2013–2017, southward currents were more frequent in the surface layer. In the bottom layer in areas deeper than 25 m, northward currents dominated in the southern part of the Archipelago Sea, while in the northern part, southward and northward currents were more evenly represented. Due to the variation in current directions, both northward and southward transports occur. During our study period, the net transport in the upper 20 m layer was southward. Below 20 m depth, the net transport was southward at the northern edge and northward at the southern edge of the Archipelago Sea. There were seasonal and inter-annual variation in the transport volumes and directions in the upper layer. Southward transport was usually largest in spring and summer months and northward transport was largest in autumn and winter months. Our results demonstrate the complexity of the transport dynamics in the Archipelago Sea. No single transect can be chosen to represent water transport through the whole area. Further studies on the water exchange processes between the Baltic proper and the Bothnian Sea through the Archipelago Sea would benefit from using a two-way nested model setup for the area.

## 1 Introduction

The Archipelago Sea is located in the Baltic Sea, between the Baltic Sea proper and the Gulf of Bothnia (Fig. 1) and together with the Åland Sea act as pathway to the water exchange between these basins. The Archipelago Sea is characterised by an extremely complex coastline bathymetry. There are over 40 000 islands and islets of various sizes. Though the mean depth of the area is only about 19 m, there are deep fault lines crossing the area in N-S or NW-SE directions that are partly deeper than 100 m.

The currents in the Archipelago Sea are strongly influenced by the complex geometry and bathymetry. Earlier measurement-based analyses of currents have shown that strong currents are occasionally induced in the deep channels (Ambjörn and Gidha-



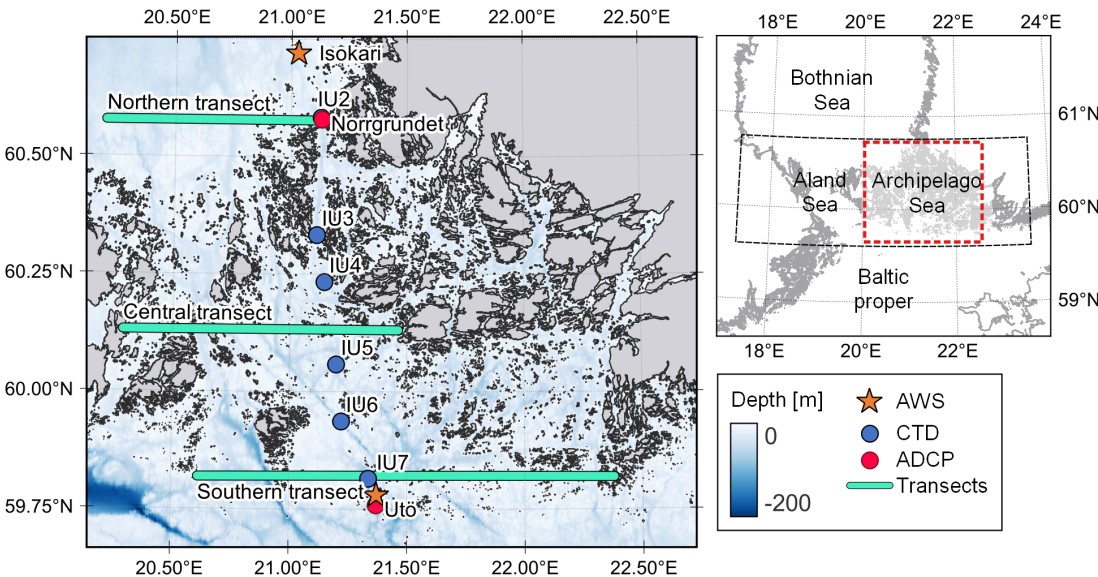

**Figure 1.** Our study area, the Archipelago Sea, and its location in the Baltic Sea (indicated with the red dashed line in the smaller map). Background color represent the bathymetry of the area (EMODnet Bathymetry Consortium, 2020). Orange stars indicate the locations of the automatic weather stations (AWS), and blue and red dots indicate the locations of the hydrography (CTD) and current (ADCP) measurements, respectively. Cyan lines indicate the locations of the transects used in volume transport calculations. The extent of the Åland Sea–Archipelago Sea model domain is indicated with black dashed line in the smaller map. (Source for the coastline: HELCOM, OpenStreetMap)

gen, 1979; Kanarik et al., 2018). The seasonal temperature stratification affects the circulation. During summer, the strongest currents are induced in the upper mixed layer and the lower part of the water column can be stagnant.

In recent years, developments in hydrodynamic modelling have improved our capabilities to describe the dynamics of com-
plex coastal areas. Depending on the type of coastal area, the required resolution and choice of methods (e.g., structured vs. unstructured grid) can vary a lot. Highest resolution (20 m–500 m) is needed in semi-enclosed basins, inlets and archipelago areas (e.g., Aleynik et al., 2016; Khangaonkar et al., 2017; Murawski et al., 2021), whereas in more open coastal areas resolution of few kilometres may be sufficient (e.g., Liu et al., 2022; Mou et al., 2022).

The earlier studies on the circulation and transport in the Archipelago Sea are based on short time-series of spatially sparse
observations (Ambjörn and Gidhagen, 1979) or on coarse-resolution models that cannot describe the complex bathymetry and dense archipelago in sufficient detail (e.g., Helminen et al., 1998; Myrberg and Andrejev, 2006). Our previous studies on modelling the Archipelago Sea have highlighted that high resolution is necessary to describe the complex geometry of this area. For example, Tuomi et al. (2018) showed that earlier coarse-resolution Baltic Sea models overestimated both the northward and southward transport of substances through the Archipelago Sea.

A recent high-resolution model study by Miettunen et al. (2020) showed that situations where substances are transported through the Archipelago Sea occur rarely. However, there is constant exchange of water between the southern Archipelago Sea





and the northern Baltic proper and between the northern Archipelago Sea and the Bothnian Sea, the southern part of the Gulf of Bothnia. Most of the time, the transport from Baltic proper and Bothnian Sea only extends to the mid-part of the Archipelago Sea suggesting that the dense archipelago performs as a 'buffer zone' where water masses are mixed.

Understanding the complex dynamics of this area and the connections to the surrounding basins are vital to marine protection and management of this vulnerable area (HELCOM, 2013). The national monitoring network is focused on the inner areas of the Archipelago Sea, closer to the mainland and there are gaps both in spatial and temporal scales. Due to this, it is not possible to capture the dynamics of this heterogeneous area with the present monitoring network (e.g., Erkkilä and Kalliola, 2007; Nylén et al., 2021). Many studies on the water properties of the Archipelago Sea area focus on specific seasons and

locations, or dedicated measurement campaigns (e.g., Suominen et al., 2010). These studies, however, do not use information on circulation, due to unavailability of suitable measured or modelled data at the time. Satellites provide information of surface layer with good spatial coverage, and they have been used to evaluate, for example, variation in turbidity (e.g. Erkkilä and Kalliola, 2004). In addition to the hydrodynamic models, also high-resolution water quality model applications have been recently developed for the Archipelago Sea, to allow more comprehensive understanding of the system (Lignell et al., 2018;

Vigouroux et al., 2019).

   To enhance our understanding on the transports within and through the Archipelago Sea we will use a new high-resolution Åland Sea–Archipelago Sea NEMO configuration that has recently been used to study transport dynamics in the Åland Sea (Westerlund et al., 2022). This is the first time that high-resolution model configuration is used to estimate volume transports in the Archipelago Sea. To support the conclusions, we use hydrographic and current measurements to evaluate the adequacy

of the model results in the Archipelago Sea. The inter-annual variation in the transport dynamics will be discussed and the accuracy of the earlier estimates will be studied.

## 2    Material and methods

### 2.1    NEMO hydrodynamic model

We used NEMO hydrodynamic model version 4.0.3 (Madec and the NEMO System Team, 2019) setup for the Åland Sea and

the Archipelago Sea (Westerlund et al., 2022). The NEMO model has previously been used in the Baltic Sea for several studies from regional (e.g., Hordoir et al., 2019; Kärnä et al., 2021) to basin-scale (e.g., Vankevich et al., 2016; Westerlund et al., 2018, 2019).

   The Åland Sea–Archipelago Sea model configuration has a horizontal resolution of 0.25 nautical mile (nmi) or approximately 500 m, and it covers the area approximately between 59.60° and 60.75° N, and 17.32° and 23.58° E (shown in Fig. 1b).

In vertical, the model uses the z* coordinate system. There are 200 levels with a level thickness of roughly 1 m down to 120 m depth, below which the thickness more rapidly increases up to 8 m at the deepest parts of the model domain.

   The bathymetry data for the model grid is compiled from two sources: VELMU (Finnish Inventory Program for the Marine Environment) bathymetry model (Finnish Environment Institute) for the Finnish EEZ and the Baltic Sea Bathymetry Database



(Baltic Sea Hydrographic Commission) for the areas outside of the Finnish EEZ. The process of bathymetric data compilation
and grid editing is described in Westerlund et al. (2022).

Meteorological forcing for the model (hourly 10 m winds, 2 m air temperature, 2 m dew point, mean sea level pressure, precipitation, snowfall rate, shortwave and longwave radiation fluxes) is taken from the ERA5 atmospheric reanalysis (Copernicus Climate Change Service; Hersbach et al., 2018). Open boundary data at the northern and southern boundaries of the model domain are compiled from the Baltic Sea Physical Reanalysis Product (CMEMS, 2022) which had a horizontal resolution of 2 nmi. Runoff data for the eight rivers that are inside the model domain is taken from VEMALA watershed model (Huttunen et al., 2016).

Westerlund et al. (2022) describe the NEMO model setup in more detail and present validation for modelled temperature, salinity and currents in the Åland Sea part of the model domain as well as for sea surface height. In this paper, we present the validation of temperature, salinity and currents for the Archipelago Sea part of the model domain.

The model simulation in this study is the same as the one used in Westerlund et al. (2022) and it covers the period from June 2012 to the end of year 2017. First half a year is regarded as an initialization period, and we analyse the results starting from the beginning of year 2013. The modelled 3D fields of temperature, salinity and currents are saved as 6 h averages and volume transports are saved once per day.

We calculated volume transports from the model results for three zonal transects (locations in Fig. 1). To calculate a time series, we integrated the volume transports over the whole transect: $F_v = \iint v \, dA = \iint v \, dz \, dl$, where $v$ is the velocity across the transect, $A$ is the area of the transect, $z$ is the depth along the transect and $l$ is the length of the transect. We also calculated the volume transport per unit length along the transect: $\int v \, dz$.

## 2.2 Observational data

### 2.2.1 Hydrography

To validate the modelled temperature and salinity, we used data from six most-visited monitoring stations in the Archipelago Sea. The stations are located along a transect directing through the outer archipelago from northern to southern edge (locations shown in Fig. 1). In total, 4–16 profiles are available per station during our modelling period (Table 1). Measurements are mostly from winter, spring and late summer, and not all the stations have been visited annually. To compare the measurements with model data, we extracted the modelled temperature and salinity profiles from the nearest representative model grid point. The model grid points are 10–66 m shallower than the measurement stations. Measured salinity was converted from psu to absolute salinity (g kg$^{-1}$) for the validation.

### 2.2.2 Currents

We used current measurements from two locations from the Archipelago Sea, from which data was available during our modelling period. One of the locations is in the northern part of the Archipelago Sea in a narrow channel (hereafter called Norrgrundet) and the other is located in the southern edge of the Archipelago Sea near the Utö islands (hereafter called



**Table 1.** Measurement stations used in model validation. N = the number of profiles from 2013–2017. Months = months that have at least one profile during these years.

| Station | N | Months | Station depth | Model depth |
|---------|----|------------------|---------------|-------------|
| IU2 | 13 | 1, 2, 5, 7, 9 | 48 m | 39 m |
| IU3 | 4 | 4, 5, 7, 9 | 50 m | 34 m |
| IU4 | 11 | 1, 2, 4, 5, 7, 9 | 50 m | 39 m |
| IU5 | 7 | 1, 4, 5, 7, 9 | 90 m | 61 m |
| IU6 | 12 | 1, 2, 4, 5, 7, 9 | 121 m | 55 m |
| IU7 | 16 | 1, 4, 5, 6, 7, 8, 9 | 93 m | 51 m |

Utö). The measurements were carried out with Teledyne RD Instruments' bottom-mounted 300 kHz WORKHORSE Sentinel Broadband Acoustic Doppler Current Profilers (ADCPs). At Norrgrundet, the bottom depth is 54 m (corresponding model grid point 39 m) and the measurements cover depths of 5 to 49 m and the period of 6 Sep 2016–16 Oct 2018. At Utö, the bottom depth is 76 m (corresponding model grid point 55 m) and the measurements cover depths of 7 to 71 m and the period of 26 Jul 2017–27 Jun 2018. In both datasets, vertical resolution is 1 m and time interval 30 min. Locations of the measurement sites are shown in Fig. 1.

### 2.2.3 Wind speed and direction

As wind is one of the most important forces inducing currents in the Archipelago Sea, we also validated the ERA5 winds against two automated weather stations (AWS), namely Utö and Isokari (locations shown in Fig. 1). Utö AWS is located in Utö island at the southern edge of the Archipelago Sea and represents open sea wind conditions from SW–SE sector. Isokari island is at the northern edge of the Archipelago Sea, and does not fully represent open sea conditions, but it gives better estimates for high winds from the W–NE sectors than the Utö AWS.

## 3 Model validation

To validate the model, we compared modelled temperature and salinity with measurements from six stations. We divided the water column into layers of 10 m and calculated the bias and RMSE for each of these layers. As the temperature and salinity profiles are sampled at 1 m intervals, the number of measurements in each layer is up to 10 times the number of profiles from which these depths were available. Down to 40 m depth, measurement data were available from all six stations and below that, only from three southernmost stations (IU5, IU6, IU7).

Similar to the results showed in our earlier study (Westerlund et al., 2022) there is bias in the salinity, with too low modelled salinities in the upper 30 m and too high in the lower layer below 40 m (Table 2). We believe this is at least partly caused by the bias in the boundary condition. Due to these biases, the haline stratification in the model is too strong. Of these six





**Table 2.** Comparison of modelled salinity and temperature against measurements from the IU stations (locations are shown in Fig. 1). N = the number of observations in each depth range.

|  |  | Salinity (g kg$^{-1}$) | | Temperature (°C) | |
| --- | --- | --- | --- | --- | --- |
| Depth range | N | Bias | RMSE | Bias | RMSE |
| 1–10 m | 630 | –0.66 | 0.75 | –0.14 | 0.84 |
| 11–20 m | 630 | –0.58 | 0.68 | 0.04 | 1.57 |
| 21–30 m | 630 | –0.20 | 0.47 | –0.74 | 1.60 |
| 31–40 m | 582 | –0.01 | 0.47 | –0.71 | 1.60 |
| 41–50 m | 350 | 0.27 | 0.55 | 0.00 | 1.65 |
| 51–60 m | 146 | 0.36 | 0.58 | –0.20 | 2.03 |

validation stations, halocline is seen only at IU7 at the southern edge of the Archipelago Sea and there is seasonal variation in its occurrence and depth (Laakso et al., 2018). The model grid is too shallow to reproduce halocline in this area. However, this does not affect our study of currents and transports, as we focus on the shallower archipelago areas with no halocline.

In modelled temperature, there was a slight underestimation at the 10 m surface layer and at the depths of 51–60 m (Table 2). The bias was largest at the depths of 21–40 m, up to –0.74 °C. RMSE increased with depth, being smallest at the surface layer and largest at 50–60 m depths. The modelled seasonal thermocline was well represented.

Direction distribution of the modelled currents is slightly narrower than that of measured currents both in Norrgrundet and Utö (Fig. 2). This is partly due to the inability of the regular model grid to represent narrow channels that are not entirely oriented in N-S or E-W direction.

In Norrgrundet, the measured currents are strongly oriented towards NNW or SSE because the measurement station is located at the mouth of a narrow channel leading towards the archipelago. The model shows similar direction distribution, but does not catch all the highest current events, with current speed of over 50 cm s$^{-1}$. Comparison shows negative bias in speed for the whole water column, being highest, –11 cm s$^{-1}$, in the bottommost layer. These underestimations can partly be addressed to the used wind forcing, which underestimates the frequency and speed of high winds (Fig. 3). The modelled currents also show too low frequency of northward currents, which could be related to ERA5 showing lower frequency of winds from S/SSE and E sectors than measurements.

In Utö, the model slightly underestimated the current speed in the upper 25 m (bias up to –1 cm s$^{-1}$), with the largest bias occurring at around 10 m depth. At the depths of 25–42 m, the model overestimated the current speed (up to 2 cm s$^{-1}$) and the largest bias occurred around 37 m depth. Below 42 m depth, bias was again negative, and largest underestimation was at the bottom layer with the bias being –5 cm s$^{-1}$. The RMSE of the current speed varied between 4–6 cm s$^{-1}$ and largest values occurred in the surface and bottom layers as well at the depth of 37 m.



Both in Norrgrundet and Utö, the bias in modelled current speed is largest at the bottommost layer. That is because the measurement points are deeper than the corresponding model grid points, so the modelled current is already slowed down by
the bottom friction at depths where the measured current does not feel the bottom yet.

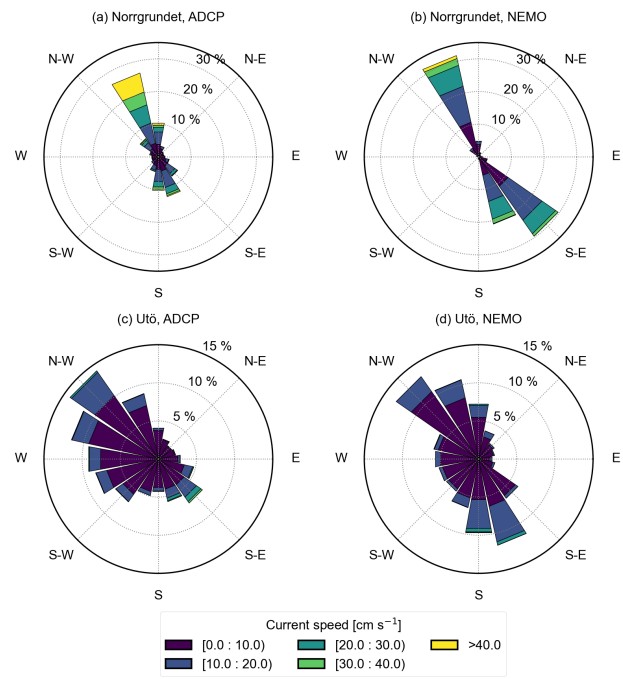

**Figure 2.** Current roses (direction to) drawn from the the measured (ADCP) and modelled currents at Norrgrundet in 6 September 2016–31 December 2017 (a, b), and at Utö in 26 July 2017–31 December 2017 (c, d). The topmost and bottommost 5 m are excluded from the model data, as they are also missing from the ADCP data.

## 4 Currents in the Archipelago Sea

The modelled currents show how the complex geometry and bathymetry affect the circulation in this area. Openness of sub-basins considerably impacts the directional distribution of the currents. In the northern part of the Archipelago Sea, the currents are strongly aligned along the N-S oriented channels, and the current directions in the 5 m surface layer alternate between north
and south (Fig. 4). Current speeds are also strongest in these narrow channels, and high current speeds exceeding 40 cm s$^{-1}$ occur frequently in the easternmost channel in the north as well as the westernmost channels south of 60.15° N. In the more open parts in the central and southern outer archipelago, currents are weaker and have a wider directional distribution, but the most frequent directions are southeast and northwest. The southward and south-eastward current directions mostly dominate throughout the model domain. An exception is the easternmost channel in the north, in which there are almost as much
northward current components as southward components.





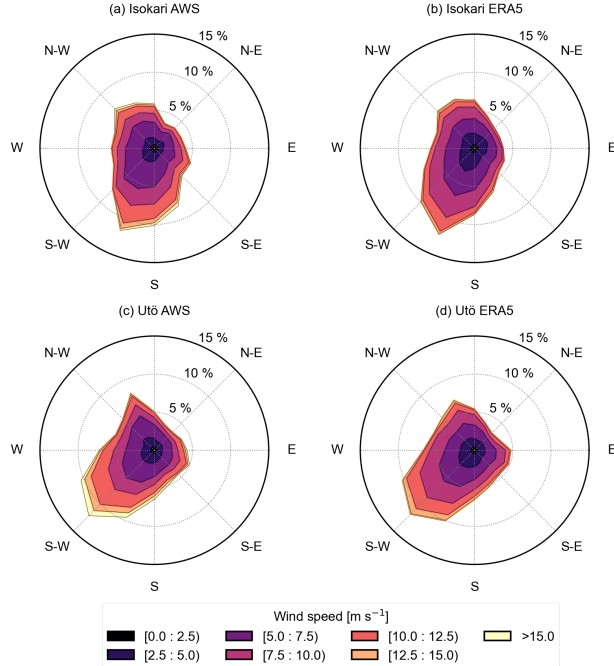

**Figure 3.** Wind roses (direction from) for years 2013–2017 drawn from Isokari (a) and Utö (c) AWS data compared to wind roses drawn from the ERA5 forcing data at the corresponding grid point (b, d).

In areas deeper than 25 m, the directional distribution of currents in the bottommost 5 m layer (Fig. 5) is strongly aligned by the bathymetry also in the southern part of the area. The northward currents dominate in the bottom layer in the southern part of the Archipelago Sea, whereas in the northern part, southward and northward currents are as common. Generally, the current speed is less than 10 cm s$^{-1}$ in the bottom layer. Currents above 10 cm s$^{-1}$ are seen in the narrow channels, and even above 20 cm s$^{-1}$ in the easternmost channel in the north and in the channel northeast from Utö.

As wind is one of the main drivers of the surface currents in the Archipelago Sea, the interannual variability in currents is strongly connected to the one in the wind conditions. During our modelling period, years 2013 and 2015–2017 resemble the long-term mean, whereas 2014 shows larger percentage of winds from eastern and south-eastern sectors than the long-term mean (Fig. 6). Therefore, also the main features of the circulation are different in 2014 than in the other years, as already demonstrated by Tuomi et al. (2018). It should be noted that due to the bi-directional nature of the currents, mean values do not reflect the circulation in the Archipelago Sea.

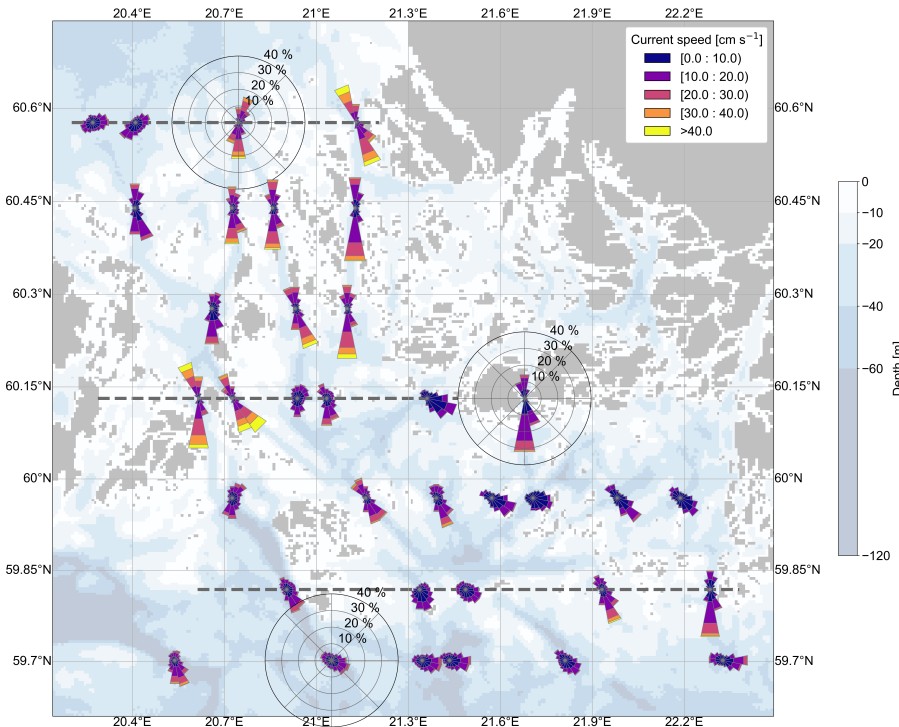

**Figure 4.** Modelled currents (direction to) in selected model grid points in the uppermost 5 m layer, 2013–2017. Each current rose shows the distribution of direction and magnitude of currents in the grid point at the centre of the rose. All current roses have the same axis limits and for clarity, the axes are drawn only for three of them. Grey dashed lines indicate the locations of the transects used in the volume transport calculations.

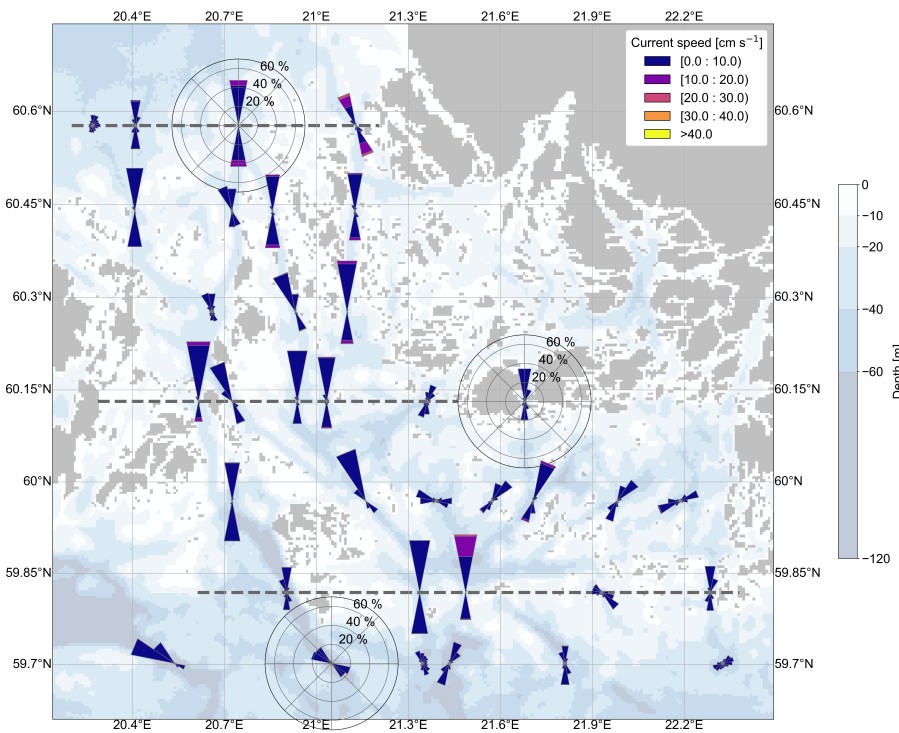

**Figure 5.** Modelled currents (direction to) in selected model grid points in the bottommost 5 m layer, 2013–2017. Each current rose shows the distribution of direction and magnitude of currents in the grid point at the centre of the rose. All current roses have the same axis limits and for clarity, the axes are drawn only for three of them. Grey dashed lines indicate the locations of the transects used in the volume transport calculations.



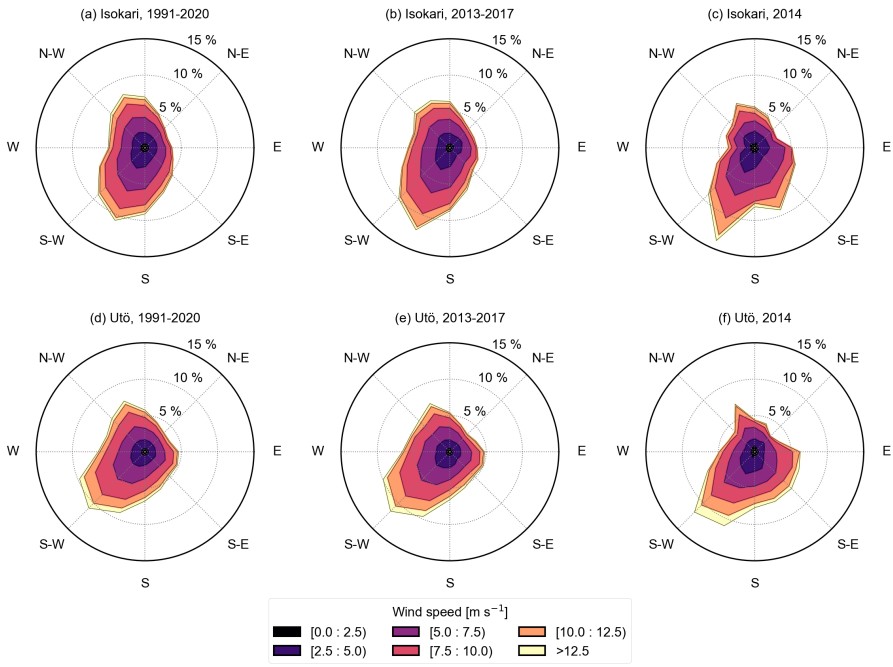

**Figure 6.** Wind roses from the ERA5 forcing data in the grid points corresponding the locations of Isokari and Utö AWSs for the 30-year period of 1991–2020 (a, d), for the years 2013–2017 (b, e), and for the year 2014 (c, f).





## 5 Volume transport in the Archipelago Sea

To study volumes and different routes of water transport in the Archipelago Sea we analysed modelled volume transports along three zonal (west-east) transects across the area. The northern and southern transects are located approximately at 60.58° N and

59.82° N, and they represent transports entering or leaving the Archipelago Sea at its northern and southern edges, respectively. The central transect is located at 60.13° N, chosen to get an estimate of transports within the area.

Transports were integrated over the whole water column, but also separately for the surface layer above 20 m depth (representing the mixed layer above the seasonal thermocline) and the lower layer below 20 m depth. As the Archipelago Sea is generally shallow and the channels that are deeper than 20 m are quite narrow, the transport in the upper layer dominates the

transport integrated over the whole water column. The shallow areas of the central Archipelago Sea also limit the lower layer transport through the area.

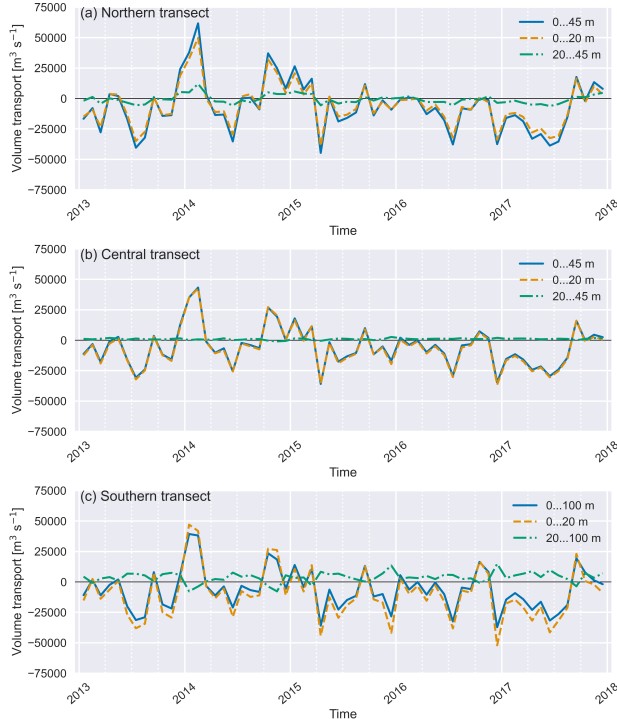

**Figure 7.** Time series of modelled monthly mean volume transport rates (in $m^3 \ s^{-1}$) through the northern (a), central (b), and southern (c) transects.

Most of the time, the monthly mean volume transport integrated along the whole transect is southward in the upper 20 m layer at all three transects (Fig. 7). The mean transport in the lower layer below 20 m is mainly southward at the northern as well as at the central transect. At the southern transect, however, the monthly mean transport in the lower layer is mainly



northward. Compared to the central and northern parts of the area, the volume transport in the lower layer is larger in the southern part of the Archipelago Sea, as the channels are deeper.

Although the net transport is southward in the surface layer, there is seasonal and annual variation on the direction and volume of the transport. Southward transport was typically largest during spring and summer months. Northward transport in the surface layer was most common in autumn, however, with considerably lower values than southward transport during other

months. An exception was the year 2014, which differs from the long-term mean in wind conditions (Sect. 4) and consequently, also in circulation and transport (Tuomi et al., 2018; Westerlund et al., 2022). Then northward transport was dominant in winter and autumn months and was of the same order of magnitude as southward transport in the other four years we studied.

To get an overview of directions and volumes and their inter-annual variability along the three transects, we calculated yearly volume transports per unit length in each model grid point along the three transects for each simulation year (not shown). In the

upper 20 m layer, the yearly net transport in 2013 and 2015–2017 was mainly southward at all the three transects. In the lower layer below 20 m, the net transport was mainly southward at the northern transect but northward at the central and southern transects. In 2014, the net transport was mainly northward both in the upper and lower layer at all the transects, but there were some channels where the net transport was southward, especially the channel east of Utö island at the southern transect and near the western edge of the northern transect. The largest transports were seen in the narrow channels of the northern

Archipelago Sea where the currents are strongest and the current direction alternates between north and south.

Because of the bi-directional currents, we wanted to study in more detail, how large are the northward and southward components that form the net transport and what is their temporal variation. To do that, we calculated monthly volume transport integrated over the whole water column, northward and southward transport separately as well as the net transport (Fig. 8). This shows that in some months with very small net transport, there are actually quite large northward and southward components

balancing each other.

The monthly variation in the transport volumes and directions is generally the same in all three transects, but the values are lower in the central and southern transects than in the northern transect (Fig. 8). There are some months when the net transport is in opposite directions in the north and south, for example in December 2014, but in those cases the absolute values are quite small.

As the Bothnian Sea has water exchange both with the Archipelago Sea and the Åland Sea, it is interesting to compare these transports at the northern edge of the Archipelago Sea with our earlier transport estimates at the northern Åland Sea (Westerlund et al., 2022). To compare, we chose one additional transect in the northern part of the Åland Sea, located at 60.3°N. This transect was also used in Westerlund et al. (2022), in which it is named "Märket South", and it represents the transport entering to or leaving from the Åland Sea at its northern boundary.

The monthly net transports in the northern Åland Sea are generally larger than in the northern Archipelago Sea, as the Åland Sea is much deeper (Fig. 9). The temporal variation in transport direction is generally similar in both transects. However, there are months when these two areas show different dynamics so that while the net transport at the northern Archipelago Sea transect is close to zero, the net transport at the Åland Sea transect is clearly northward (October 2016) or southward (e.g., August 2014, January–February 2016 and October 2017).



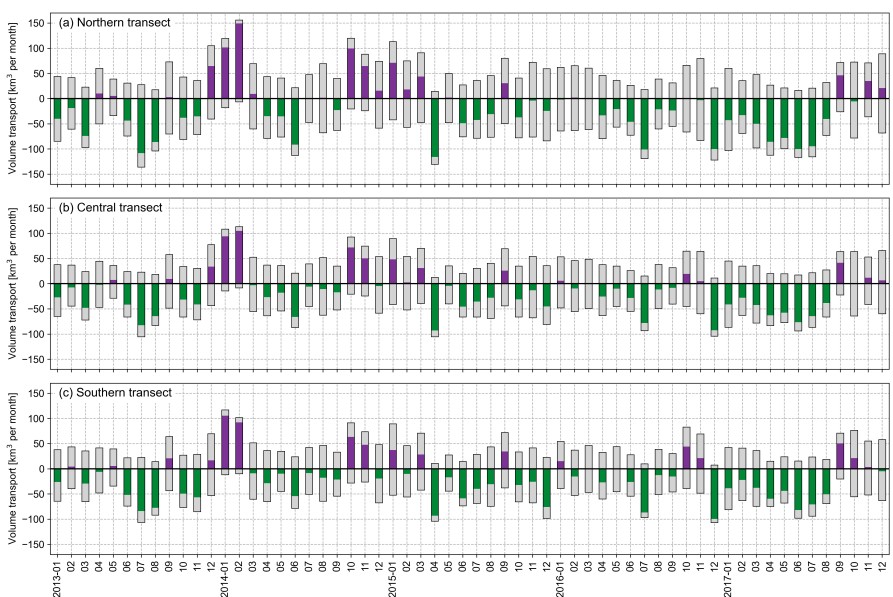

**Figure 8.** Monthly transport (km$^3$ per month) in the whole water column in the northern (a), central (b), and southern (c) Archipelago Sea. Gray bars indicate the northward (positive) and southward (negative) monthly transport separately and the purple and green bars indicate the net transport.

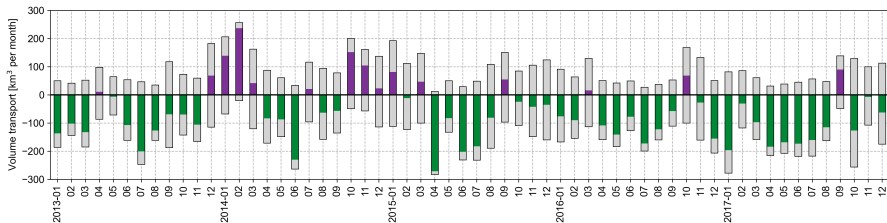

**Figure 9.** Monthly transport (km$^3$ per month) in the northern Åland Sea in the whole water column. Gray bars indicate the northward (positive) and southward (negative) monthly transport separately and the purple and green bars indicate the net transport. Note the different y axis limits compared to Fig. 8.

Averaging over our study period of 2013–2017, the mean of the net transport over the northern Archipelago Sea transect is –206 km$^3$ per year. This is approximately 28 % of the mean net transport over the Åland Sea transect which was –745 km$^3$ per year. While the net transports in the Åland Sea are generally larger than in the Archipelago Sea, in the year 2014 the net northward transport at the northern edge of the Archipelago Sea was 1.3 times larger than at the northern edge of the Åland Sea. If the year 2014 would be left out when averaging, there would be less cases of northward transport and the resulting mean net transport towards south would be –321 km$^3$ per year at the northern Archipelago Sea transect and –979 km$^3$ per year at the Åland Sea transect.



## 6 Discussion

Our results show that the net transport in the surface mixed layer is southward during the study period. This is expected due to the voluminous river discharges to the Gulf of Bothnia flowing out of the gulf through the Åland Sea and the Archipelago Sea. The earlier estimates of transports in this area that have suggested the opposite, that is, northward net transport through the Archipelago Sea (Ambjörn and Gidhagen, 1979; Helminen et al., 1998). However, there are months of northward net transport also during our study period (especially during winters 2013–2014 and 2014–2015).

Ambjörn and Gidhagen (1979) estimated the monthly volume transports through the Åland Sea and the Archipelago Sea in June–August 1977 based on current measurements in five locations in the northern Åland Sea and three locations in the Archipelago Sea (Table 3). For June, the estimated net transports were small and southward in both areas, whereas for July and August, the net transport was southward in the Åland Sea and northward in the Archipelago Sea.

**Table 3.** Monthly volume transports (in km$^3$ per month) from Ambjörn and Gidhagen (1979).

|  | Åland Sea | | | Archipelago Sea | | |
|---|---|---|---|---|---|---|
|  | North | South | Net | North | South | Net |
| June 1977 | 63 | –78 | –15 | 28 | –31 | –3 |
| July 1977 | 49 | –136 | –87 | 44 | –25 | 19 |
| August 1977 | 46 | –135 | –89 | 36 | –14 | 22 |

Compared to their estimates, the order of magnitude of our modelled monthly net transports are similar in both areas. However, their estimate suggests northward net transport in the Archipelago Sea, whereas our model shows southward net transport. The transport estimate for the Archipelago Sea by Ambjörn and Gidhagen (1979) are based on short measurement time series limited to thermally stratified season in 1977. Compared to the long-term wind conditions in Utö AWS, the most frequent directions during this period were NNW, S and E, and the SW winds typical for the area were mostly missing. Thus, their transport estimates represent a specific summer season not representative for the long-term annual mean conditions in the area.

When studying transports in a complex area like the Archipelago Sea, the locations of the transects must be chosen carefully so that they represent what they are intended to. The estimates of Ambjörn and Gidhagen (1979), for the Archipelago Sea, are based on measurements on three locations, chosen to represent the three deepest and largest channels crossing area. As one is located at the northern edge of the area and two in the central part of the area, they represent different parts of the archipelago. Given the complexity of the archipelago and the differences in the modelled current direction distribution between the different parts of the area, it is not possible to use single transect to describe transport through the whole Archipelago Sea from the Baltic proper to the Bothnian Sea or vice versa. Also, based on our earlier studies, the transport through the area rarely occurs during a single event and requires considerable amount of time (Miettunen et al., 2020). In this study, the northern transect represents



the water exchange between the Bothnian Sea and the Archipelago Sea, and the southern transect between the Archipelago Sea and the Baltic proper.

Our simulation period, five years, is relatively short. But with the high-resolution model setup, the computational cost is heavy and making long hindcasts a challenge. As the surface currents are mainly wind-driven, the interannual variation in currents and transports is related to the interannual variation in wind conditions. To generalize our results, we compared the wind conditions during these five years against the long term mean wind conditions. During the years with winds that resemble the long-term wind conditions with dominant SW winds, the net transport in the Archipelago Sea is southward. However, exceptions occur, like the year 2014 that had larger contribution of SE and NNW winds than on average and showed northward net transport. Seasonal variation in transports is also related to the wind conditions but also to the freshwater discharge to the Gulf of Bothnia. The river discharges are largest in spring, and southward net transport is usually strongest in spring and summer, when the winds are usually weaker. In autumn and early winter, the winds are stronger and there are fewer northerly winds than in spring or summer, and the volume of northward transport is largest. To capture these interannual and seasonal variations in transport directions, several years has to be modelled. Although longer period would reduce the uncertainties in the transport estimates, we consider our modelling period to be a good representation of overall dynamics in this area.

Modelling in this complex archipelago area is challenging and requires high resolution and sufficiently good bathymetric data as shown in our earlier studies (Tuomi et al., 2018; Miettunen et al., 2020). High-resolution model data enables us to gain more comprehensive understanding of circulation and transport dynamics in this area than the earlier estimates presented, for example, by Ambjörn and Gidhagen (1979) and Helminen et al. (1998).

Possibility to evaluate the model results against recent current measurements demonstrated the model's capability to describe the circulation dynamics in the area. However, uncertainties remain regarding, for example, the bathymetric data (Westerlund et al., 2022), the grid resolution (Tuomi et al., 2018), and the accuracy of both the lateral boundary conditions and the atmospheric forcing data. Reanalysis data used as the open sea boundary conditions are daily averages, meaning that the phenomena with smaller timescales (such as seiches) cannot be included. As noted already by Westerlund et al. (2022), one way to address the issues caused by the boundary conditions would be to develop a two-way nested configuration with a coarse resolution Baltic Sea model and the high-resolution local model.

With a two-way nested model setup, the dynamics of this buffer zone between the Baltic Proper and the Gulf of Bothnia could be studied more comprehensively. It could also be used to study dynamics that affect the state of the Bothnian Sea: for example, at which depths and along which route nutrient-rich water can flow to the Bothnian Sea and when this kind of water transport occurs. Further, we could study if there has been changes in the water exchange dynamics which would in part explain the changes that have been recently observed in the state of the Bothnian Sea (e.g. Kuosa et al., 2017; Polyakov et al., 2022). We could also study the exchange between the Archipelago Sea and the Baltic Proper and the Bothnian Sea, and how these open sea areas affect the coastal waters, for example in the terms of nutrient loads.



## 7 Conclusions

This is the first time that transports in the Archipelago Sea and between the Archipelago Sea and the Bothnian Sea have been studied using a high-resolution coastal modelling system. The five-year simulation period and good spatial coverage allow a detailed evaluation of the circulation and transport dynamics.

Our analysis showed that the currents are steered by the geometry of the islands and straits and the bottom topography. In the narrow channels, current directions alternated between south and north in the northern part of the Archipelago Sea and
between south-east and north-west in the southern part. More open areas in the southern part of Archipelago Sea showed wider directional distribution of currents. In the surface layer, southward and south-eastward currents were slightly more frequent. In the bottom layer in the areas deeper than 25 m, northward currents dominate in the southern part of the area, whereas in the northern part, southward and northward currents are more evenly represented.

Due to the bi-directional currents, both northward and southward water transport occur in the Archipelago Sea. The south-
ward flow is more frequent in the surface layer, though in suitable wind conditions, water flow from south to north through the whole area is possible. This is partly due to the voluminous freshwater discharge to the Gulf of Bothnia leading to southward net volume transport in the 20 m upper layer. Southward transport is usually largest in spring and summer months. Northward transport is largest in autumn and winter months. In the lower layer, below 20 m depth, the net transport is southward in the northern part of the Archipelago Sea and northward in the southern part.

The net volume transports in the Archipelago Sea are smaller than in the Åland Sea because it is much shallower. During our simulation period, the mean yearly net transport at the northern edge of the Archipelago Sea is approximately one third of that at the northern edge of the Åland Sea. The transport dynamics in the Archipelago Sea are more complex than in the Åland Sea. For example, there are months when the direction of the net transport is close to zero at the northern edge of Archipelago Sea, while at the northern edge of the Åland Sea, the net transport is clearly northward or southward. In general, however, the
monthly variation in the direction of the net transports is similar in both areas.

These transport estimates demonstrate the nature of the Archipelago Sea as a buffer zone between the Baltic Proper and the Bothnian Sea. The transport dynamics in the Archipelago Sea are complicated so that no single transect can be chosen to represent the transport through the whole area. More comprehensive studies on the role of the Archipelago Sea in the water exchange processes between the Baltic Proper and the Bothnian Sea are needed and would benefit from using a two-way nested
model setup with the high-resolution Åland Sea–Archipelago Sea model and the coarser-resolution Baltic Sea model.

*Code and data availability.* The standard NEMO model source code is available from the NEMO web site at https://www.nemo-ocean.eu/, https://doi.org/10.5281/zenodo.3878122 (Madec and the NEMO System Team, 2019). The NEMO configuration files for the Åland Sea and Archipelago Sea setup are available from https://github.com/fmidev/nemo-archs (Westerlund and Miettunen, 2021). Baltic Sea reanalysis data (CMEMS, 2022) used as model boundary data are no longer available from the Copernicus Marine Service because the reanalysis has
been replaced by a newer product. Atmospheric forcing data are available from the Copernicus Climate Service at https://doi.org/10.24381/ cds.adbb2d47 (Hersbach et al., 2018). The bathymetric input file for the Åland Sea and Archipelago Sea NEMO configuration is not available



due to current Finnish Environment Institute (SYKE) policy regarding the VELMU bathymetric data. River runoff data are available from SYKE on request. The ADCP datasets for the Norrgrundet and Utö stations and the CTD dataset for the IU stations are available on request from the authors. This study uses data from the Baltic Sea Bathymetry Database (Baltic Sea Hydrographic Commission, 2013) version 0.9.3,

downloaded from http://data.bshc.pro/ (last access: 24 July 2018). This study has been conducted using EU Copernicus Marine Service Information. Contains modified Copernicus Climate Change Service information 2020. Neither the European Commission nor ECMWF is responsible for any use that may be made of the Copernicus information or data it contains.

*Author contributions.* AW carried out the model simulations. EM performed the analyses of the model results with contribution from LT and KM. EM validated the model results and and performed visualization with contribution from HK. All authors took part in the discussion of

the results. EM and LT prepared the manuscript with contributions from all authors.

*Competing interests.* The authors declare that they have no conflict of interest.

*Acknowledgements.* This work has been partially funded by the Strategic Research Council at the Academy of Finland (contract no. 312650, BlueAdapt), and the Finnish Ministry for the Environment Water Protection Programme 2019–2023.



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
