# Peer review of "Transport dynamics in a complex coastal archipelago"

_EGUsphere, 2023_

## Referee Comment (RC1)

**Transport dynamics in a complex coastal archipelago**

The MS is aimed to study the currents and volume transports of water in the Archipelago Sea using results from the high-resolution (0.25 nautical miles) 3D NEMO ocean model. From the same model experiments, results about the neighboring Åland Sea have been published by Westerlund et al. (2022). Model results in the Archipelago Sea are validated with acceptable results by the observations of temperature, salinity and currents using the data from the archives. Results part of the MS present (chapter 4) statistics of modelled currents in relation to winds, using mainly directional "roses", and (chapter 5) time series of monthly mean volume transports across the selected 3 west-east transects. In particular, the study reveals that currents are steered by the geometry of the islands and straits and the bottom topography. Net transport in the upper 20 m layer was southward. Monthly volume transport had maximum southward direction in spring and northward direction in autumn and winter. Thus, the study and its results are generally interesting and could be published.

In the following, I elaborate the background in order to give recommendations how the MS might be made more significant and interesting.

A. The modelling results of the Archipelago Sea with a grid step of 0.25 nautical miles have been published earlier by Tuomi et al. (2018) and Miettunen et al. (2020). They used the 3D COHERENS model. Present MS should also reference to the earlier model. It should be interesting to know is there an improvement from COHERENS to NEMO. Oceanographic results of the two earlier studies are referenced in the Introduction. In my understanding, the oceanographic results of present MS does not go much beyond these earlier studies. Perhaps this feeling is subjective and fed by massive use of the term "complex" (15 cases vs 2 and 6 cases in earlier studies). Authors are encouraged to look how to include more oceanography and reduce information-poor terms/formulations like "complex" and "resolution" (26 cases).

B. The study reveals steering of currents along closely spaced isobaths in straits, channels and trenches. This issue of topographically constrained currents is generally known and could be more presented and discussed, including more references to the theoretical studies and observations in nearby Baltic regions. Is the role of islands to guide the flow, without significant frictional slow-down?

C. Complementary to the current roses in Figs. 4-5, it should be interesting to see (seasonally?) mean current maps (perhaps together with persistency contours).

D. Section of volume transports is interesting, but more information on water budget, transect areas and forcing factors could be presented. (a) Time series of monthly mean transports as shown in Fig. 7 are similar on surface and intermediated layers. Nevertheless, they are also similar on different transects, with correlation above 0.9 (I made this check). This indicates large-scale forcing of volume transport. (b) Forcing of volume transports has been discussed but not evaluated. Ambjörn and Gidhagen (1979) have concluded: "Main driving force on the net current, when the vertical stratification is weak, is the surface slope along the channel. Local acceleration and bottom friction are also important." This can be directly evaluated from the monthly mean model results. It should also be interesting to know what wind stress projections (to what angle) cause sea level slopes across the Archipelago Sea favoring northward or southward flows. For example, SE and NNW winds in 2014 created larger volume transports than in other years; was it related to the larger sea level slopes?

E. The MS emphasizes further need to increase the resolution. How many details are reasonable? Thousands of islands are making already some statistical entity. For example, flows in the porous media (e.g. Pratt, L.J. and Spall, M.A., 2003. A porous-medium theory for barotropic flow through ridges and archipelagos. Journal of physical oceanography, 33(12), pp.2702-2718.) can be modelled without counting each individual grain and/or pore. (Consider also Darcy law).

F. The title of MS is too general for the present content. Archipelago dynamics in general oceanographic sense is not presented and discussed. References to the other archipelago sea studies focus mainly on technical details, such as model setup, need for higher model resolution and more dense monitoring network. I recommend to rephrase the title.

I include also some minor remarks.

1) The term "high-resolution" (counted 11 times) could be specified.
2) The term "area" is used as a synonym for "region". It could be useful to present and discuss actual geometrical areas of the transects, hypsographic curves of the regions etc.
3) Line 4: It has to be specified what NEMO is, even in the abstract (an oceanographic model?)
4) Lines 40-44 say that "situations where substances are transported through the Archipelago Sea occur rarely" and "there is constant exchange of water". How water exchange occurs without transport of substances? The role of salt exchange is not figured out, although there should be long-term salt flux based on the Knudsen formulae.
5) Lines 78-79: open boundary data were taken from the Baltic Sea Physical Reanalysis Product. This data set has daily mean values for currents, temperature and salinity. How the boundary values with periods shorter than a day were taken into account? The reader could be interested to understand the main features without reading Westerlund et al. (2022).
6) Line 87: "temperature, salinity and currents are saved as 6 h averages" is nearly able to cover the daily cycle. How shorter period processes like 14-hour inertial oscillations, sea level variations can be taken into account? // Considered in the discussion, lines 273-276 but could be brought in earlier, in the methods.
7) Lines 128-129: "The model grid is too shallow to reproduce halocline in this area. However, this does not affect our study of currents and transports, as we focus on the shallower archipelago areas with no halocline." It should be better justified. For example, presenting the fraction of halocline-covered area to the area of whole transect.
8) Lines 306-307: the statement "Archipelago Sea as a buffer zone between the Baltic Proper and the Bothnian Sea" needs explanation. It cannot be directly deduced from the synchronous monthly mean transports presented in Fig. 7. Regarding spreading of tracers, indeed Miettunen et al. (2020) have shown by integration of Lagrangian transport that "only a small percentage of the particles released in the southern and northern parts of the model area entered the middle and inner archipelagos." Perhaps the flow speed corresponding to monthly mean transport is so small that water cannot be transported through the all sections during a month. Distance between northern and southern transects is about 85 km, there should be the speed 3.3 cm/s to cover such distance.
9) Lines 307-308: there is a statement "The transport dynamics in the Archipelago Sea are complicated so that no single transect can be chosen to represent the transport through the whole area." On the other hand, Fig. 7 shows that monthly transports across the three sections are similar. There seems to be some controversy; please explain in the revised text.
10) Figures 4 and 5 could be combined together as (a) and (b) since their only difference is in the selection of layer: uppermost 5 m vs bottommost 5 m.
11) Figures 3 and 6 are very similar and contain repeated information. Perhaps to keep only one figure.
12) Net transport in Fig. 8 is the same as already presented in Fig. 7. Please try to avoid duplication.
13) Figures 8 and 9 contain the same information, only for the two different transects. By such presentation, comparison of transports is not straightforward. Please consider some other reader-friendly presentation.

I recommend an editorial revision of the MS.

---

## Community Comment (CC1)

This review reflects comments and contributions by Júlia Sambugaro and Maria Carolina Matos resulted from the graduate-level course "*How to Read and Evaluate Scientific Papers and Preprints*" from the University of São Paulo, which aimed to provide students the opportunity to review scientific articles, develop critical and constructive discussions on the endless frontiers of knowledge, and understand the peer review process.

The preprint examines circulation and water transport in the Archipelago Sea, Finland. Earlier works in the study area failed to estimate the water transport due to the complex topography of the archipelago, leading to an overestimation. The vulnerability of this region justify the necessity of understanding the transport dynamics.

Using a high-resolution NEMO configuration, the study reveals that currents are influenced by the area's geography, resulting in stronger currents in narrow channels and weaker currents in more open spaces. Seasonal and interannual variations in transport volume and direction were observed, emphasizing the intricate dynamics of the Archipelago Sea's water exchange processes.

The work rectifies past limitations and offers valuable insights for managing this unique marine ecosystem, presenting an advancement in understanding transport dynamics in the Archipelago Sea.

**MAJOR COMMENTS**

**Introduction**

- Consider adding a concise statement about the main research goal to improve the last paragraph that previews of what the preprint aims to achieve.

- Your study group has been conducting various research projects in the region, contributing to a better understanding of such a complex area. We believe that in order to reach broader audiences, it would be beneficial to include a map depicting a larger surrounding area, along with a reference to the corresponding country in the text (beginning of the Introduction). With this addition, the current Figure 1 in the introduction could be relocated to the Methods section, where it would fit more appropriately.

**Methods**

- There is a concern regarding observing seasonal variations through a model validated using measurements that don't have complete seasonal data coverage. Maybe in the future, collect data seasonally would bring more reliability to modeled seasonal variations.

**Model validation**

- Another concern revolves around the differences in depth between what was measured and modeled. Despite the RMSE being lowest for the first ten meters, the magnitude of the value is high enough to raise doubts about the ability of the model to predict data.

- It was mentioned that *"Direction distribution of the modeled currents is slightly narrower than that of measured currents both in Norrgrundet and Utö (Fig. 2)"*, perhaps the bigger discrepancy between the current speed and direction should also be mentioned.

**Currents in the Archipelago Sea**

- Very interesting that wind can be represented by mean values while currents can't due to its bi-directional nature.

**Discussion**

- To show the interannual variation was presented the wind and its variability data. Therefore to show the seasonal variation the importance of wind and fluvial discharge was mentioned. So, it would be helpful to provide the fluvial discharge data.

- It is great that you recognize the uncertainties of the model and already suggest the next improvement in the method: *"As noted already by Westerlund et al. (2022), one way to address the issues caused by the boundary conditions would be to develop a two-way nested configuration with a coarse resolution Baltic Sea model and the high-resolution local model."*

**MINOR COMMENTS**

- It is mentioned in the text "Fig. 1b", but in the figure itself the maps are not named "a" or "b".

- The smaller map in Figure 1 could be improved by coloring the land masses another color to make it distinct from the sea area.

---

## Author Comment (AC1)

**Author's reply to referee comments**

Miettunen, E., Tuomi, L., Westerlund, A., Kanarik, H., and Myrberg, K.: Transport dynamics in a complex coastal archipelago, EGUsphere [preprint], https://doi.org/10.5194/egusphere-2023-1547, 2023.

Below, referee comments are displayed with italic font, highlighted with a grey background. Author replies are without highlighting. When listing the changes in the revised manuscript, line numbers refer to the "Author's tracked changes" file.

**RC1, Anonymous Referee #1, 30 Aug 2023**

> *Review of MS by Elina Miettunen et al.*

> *Transport dynamics in a complex coastal archipelago*

> *The MS is aimed to study the currents and volume transports of water in the Archipelago Sea using results from the high-resolution (0.25 nautical miles) 3D NEMO ocean model. From the same model experiments, results about the neighboring Åland Sea have been published by Westerlund et al. (2022). Model results in the Archipelago Sea are validated with acceptable results by the observations of temperature, salinity and currents using the data from the archives. Results part of the MS present (chapter 4) statistics of modelled currents in relation to winds, using mainly directional "roses", and (chapter 5) time series of monthly mean volume transports across the selected 3 west-east transects. In particular, the study reveals that currents are steered by the geometry of the islands and straits and the bottom topography. Net transport in the upper 20 m layer was southward. Monthly volume transport had maximum southward direction in spring and northward direction in autumn and winter. Thus, the study and its results are generally interesting and could be published.*

> *In the following, I elaborate the background in order to give recommendations how the MS might be made more significant and interesting.*

We would like to thank the referee for taking the time to review the manuscript and providing recommendations on how to make it more significant and interesting. We greatly appreciate the feedback and think that the manuscript is now improved after addressing these comments. Please find our replies to the comments below.

> *A. The modelling results of the Archipelago Sea with a grid step of 0.25 nautical miles have been published earlier by Tuomi et al. (2018) and Miettunen et al. (2020). They used the 3D COHERENS model. Present MS should also reference to the earlier model. It should be interesting to know is there an improvement from COHERENS to NEMO. Oceanographic results of the two earlier studies are referenced in the Introduction. In my understanding, the oceanographic results of present MS does not go much beyond these earlier studies. Perhaps this feeling is subjective and fed by massive use of the term "complex" (15 cases vs 2 and 6 cases in earlier studies). Authors are encouraged to look how to include more oceanography and reduce information-poor terms/formulations like "complex" and "resolution" (26 cases).*

Thank you for this comment. We agree and have added reference to the earlier COHERENS-based model setup to the introduction (lines 74–76 in the "Author's tracked changes" file) and discussion (lines 327–329) sections.

Thank you for pointing out our excessive use of the term "complex". In the revised manuscript, we use more diverse language and more descriptive formulations (e.g., lines 36–37, 316–317). Regarding the resolution, we have included definitions of what we mean when talking about "high" and "coarse" resolution models (lines 45–48).

> B. The study reveals steering of currents along closely spaced isobaths in straits, channels and trenches. This issue of topographically constrained currents is generally known and could be more presented and discussed, including more references to the theoretical studies and observations in nearby Baltic regions. Is the role of islands to guide the flow, without significant frictional slow-down?

Topographically restricted currents are quite common in the Baltic Sea due to its varying topography, elongated gulfs and archipelagos. We think that theoretical discussion on this is out of scope of this manuscript. However, we have added a bit more information about this to the introduction, concentrating on the specific features of the Archipelago Sea where the flow is guided both by the islands and bathymetry (lines 26–33).

> C. Complementary to the current roses in Figs. 4-5, it should be interesting to see (seasonally?) mean current maps (perhaps together with persistency contours).

When studying circulation in the model region, we have analysed the monthly and seasonal currents by drawing both mean current maps and current roses. The bi-modal nature of the currents, especially in the northern and central parts of the area, makes the current means misleading. While for some periods, the mean current field can actually represent the most dominant current direction, in some cases the mean does not show either of the dominant directions. This is especially the case for autumn: the mean current field calculated from the vector components shows eastward currents also in the northern part of the region where the roses show clearly that northward and southward directions dominate (see the figure below). That is why we have decided to present the circulation with current roses only.

[Figure]

Figure: Current roses for selected grid points (left) and mean current field (right) for the 5 m surface layer in autumn (Oct–Dec), 2013–2017.

> *D. Section of volume transports is interesting, but more information on water budget, transect areas and forcing factors could be presented. (a) Time series of monthly mean transports as shown in Fig. 7 are similar on surface and intermediated layers. Nevertheless, they are also similar on different transects, with correlation above 0.9 (I made this check). This indicates large-scale forcing of volume transport. (b) Forcing of volume transports has been discussed but not evaluated. Ambjörn and Gidhagen (1979) have concluded: "Main driving force on the net current, when the vertical stratification is weak, is the surface slope along the channel. Local acceleration and bottom friction are also important." This can be directly evaluated from the monthly mean model results. It should also be interesting to know what wind stress projections (to what angle) cause sea level slopes across the Archipelago Sea favoring northward or southward flows. For example, SE and NNW winds in 2014 created larger volume transports than in other years; was it related to the larger sea level slopes?*

(a) It is true that the surface layer transports are similar at all the three transects. However, the lower layer transports are different at different transects. This was not perhaps clearly visible in the original figures. In response to the 12th minor comment about duplication of data in Figs. 7 and 8, we have replaced those figures in the revised manuscript. The new ones show the differences between the upper and lower layer at the different transects more clearly.

(b) This is the first time that this kind of model analysis of volume transports is conducted for the Archipelago Sea. We chose the same method used in Westerlund et al. (2022) in the Åland Sea so that we can compare the results for these two regions. We agree that the water budget and other forcing factors are important to analyse to better understand the dynamics of these regions and the water exchange between the different basins. However, as both the Åland Sea and the Archipelago sea act as a pathway for the water exchange between the Baltic proper and the Bothnian Sea, the analysis of the forcing should include both areas. This kind of analysis is out of scope of this manuscript and will be continued in the future. In our future work, we aim to use longer model simulations, preferably using a two-way nested model setup, and also include measurements conducted in the region during the past years.

> *E. The MS emphasizes further need to increase the resolution. How many details are reasonable? Thousands of islands are making already some statistical entity. For example, flows in the porous media (e.g. Pratt, L.J. and Spall, M.A., 2003. A porous-medium theory for barotropic flow through ridges and archipelagos. Journal of physical oceanography, 33(12), pp.2702-2718.) can be modelled without counting each individual grain and/or pore. (Consider also Darcy law).*

We think that the current model resolution of 0.25 nautical miles is enough for our studies. In general, the resolution is enough to describe the main waterways in sufficient detail. However, some of the channels are quite narrow and oriented diagonally with respect to the model grid. Higher grid resolution could improve the description of those channels and their orientation and thus improve especially the estimates for the lower layer transports.

Impression that further increase in resolution would be needed for this study was not intended. We have rephrased the parts that might have given this impression to avoid misinterpretation (e.g., lines 324, 338).

> *F. The title of MS is too general for the present content. Archipelago dynamics in general oceanographic sense is not presented and discussed. References to the other archipelago sea studies focus mainly on technical details, such as model setup, need for higher model resolution and more dense monitoring network. I recommend to rephrase the title.*

We chose to use a more generic title for this manuscript, since although these types of archipelago areas are rare, they still exist also outside our study area. We also think that our study and the methods we use can be of interest also to researchers focusing on other coastal archipelago regions in and outside the Baltic Sea.

> *I include also some minor remarks.*

> *1) The term "high-resolution" (counted 11 times) could be specified.*

It is true that this term is a bit vague and it depends on the modelled region whether a model resolution can be considered high or not. As mentioned in our reply to the comment A, we have now specified in the revised manuscript what we mean when talking about high and coarse resolution models (in the introduction, lines 45–48).

> *2) The term "area" is used as a synonym for "region". It could be useful to present and discuss actual geometrical areas of the transects, hypsographic curves of the regions etc.*

We included plots of the topography along the transects in Fig. 1. These show how the thickness of the upper and lower layers vary along the transects and how the lower layer is mostly present only in the channels crossing the area.

> *3) Line 4: It has to be specified what NEMO is, even in the abstract (an oceanographic model?)*

This is now specified in the abstract (lines 4–5) and in the introduction (line 71).

> *4) Lines 40-44 say that "situations where substances are transported through the Archipelago Sea occur rarely" and "there is constant exchange of water". How water exchange occurs without transport of substances? The role of salt exchange is not figured out, although there should be long-term salt flux based on the Knudsen formulae.*

These sentences were perhaps not clear in the original manuscript. What we mean is that there is constant water exchange between the outer parts of the northern and southern Archipelago Sea with their neighbouring basins (Bothnian Sea in the north and Baltic Proper in the south). However, there is not constant water exchange between the Baltic Proper and the Bothnian Sea through the Archipelago Sea because the exchange is limited by the archipelago. We have now rephrased this part in the revised manuscript so that it is more clear (lines 50–57).

> *5) Lines 78-79: open boundary data were taken from the Baltic Sea Physical Reanalysis Product. This data set has daily mean values for currents, temperature and salinity. How the boundary values with periods shorter than a day were taken into account? The reader could be interested to understand the main features without reading Westerlund et al. (2022).*

Currents, temperature and salinity in the open boundary data are indeed daily mean values, but sea surface heights are hourly values. This was not mentioned in the model description in the original manuscript but it is now included in the revised version (lines 100–101). Thank you for pointing this out.

*> 6) Line 87: "temperature, salinity and currents are saved as 6 h averages" is nearly able to cover the daily cycle. How shorter period processes like 14-hour inertial oscillations, sea level variations can be taken into account? // Considered in the discussion, lines 273-276 but could be brought in earlier, in the methods.*

The model results are indeed saved as 6 h averages, but the model of course accounts for short-term variability. As we analyse the transports as monthly sums and net values, we see that the output frequency is enough for this study.

Modelled sea level was saved at 1 h intervals and validated in our earlier paper (Westerlund et al., 2022) using tide gauge data from Föglö station which is located in the western Archipelago Sea. This validation was not clearly referenced in the original manuscript. In the revised manuscript, we now summarise the sea level validation done in Westerlund et al. (2022) at the beginning of Section 3 (lines 140–143).

*> 7) Lines 128-129: "The model grid is too shallow to reproduce halocline in this area. However, this does not affect our study of currents and transports, as we focus on the shallower archipelago areas with no halocline." It should be better justified. For example, presenting the fraction of halocline-covered area to the area of whole transect.*

Halocline is seen only at the southern edge of the Archipelago Sea where the IU7 measurement station is located. The other IU stations north of this show no halocline, not even the closest one, IU6, that is only 15 km away from IU7, and is 121 m deep. So we argue that even though the model cannot reproduce the occasional halocline seen in IU7, this has no effect on the main analysis of our study, as there is no halocline northward from this and thus possible sub-halocline transports would not be able to propagate northward. We have modified this part in the revised manuscript so that it is more clear what we mean (lines 155–159).

*> 8) Lines 306-307: the statement "Archipelago Sea as a buffer zone between the Baltic Proper and the Bothnian Sea" needs explanation. It cannot be directly deduced from the synchronous monthly mean transports presented in Fig. 7. Regarding spreading of tracers, indeed Miettunen et al. (2020) have shown by integration of Lagrangian transport that "only a small percentage of the particles released in the southern and northern parts of the model area entered the middle and inner archipelagos." Perhaps the flow speed corresponding to monthly mean transport is so small that water cannot be transported through the all sections during a month. Distance between northern and southern transects is about 85 km, there should be the speed 3.3 cm/s to cover such distance.*

We agree that our statement about the Archipelago Sea being a buffer zone cannot be deduced from the results presented in this manuscript. Thank you for pointing this out. In addition to the work done in this paper, we base the statement on the earlier work done in this area by ourselves and others. We have modified the conclusions accordingly (lines 370–371).

*> 9) Lines 307-308: there is a statement "The transport dynamics in the Archipelago Sea are complicated so that no single transect can be chosen to represent the transport through the whole area." On the other hand, Fig. 7 shows that monthly transports across the three sections are similar. There seems to be some controversy; please explain in the revised text.*

It is true that the monthly transports across the three transects are similar when looking at the surface layer or the whole water column, and thus the dominant southward net transport is similar everywhere. However, transports in the lower layer are generally towards opposite directions in

north and south. We have edited the discussion part in the revised manuscript to explain this (lines 297–306) and also rephrased the abstract (lines 16–17) the conclusions (lines 371–372) accordingly.

> 10) Figures 4 and 5 could be combined together as (a) and (b) since their only difference is in the selection of layer: uppermost 5 m vs bottommost 5 m.

This is a good idea. We will discuss this with the technical editor if the manuscript is accepted for publication.

> 11) Figures 3 and 6 are very similar and contain repeated information. Perhaps to keep only one figure.

We decided to keep both figures but modified Fig. 6 to avoid repetition. Fig. 3 is part of validation, showing the differences between the wind forcing and observations. The new version of Fig. 6 now shows the wind roses for the 30-year period as well as for the years 2013–2017 separately, to show how 2014 differs from the other years and from the long-term wind distribution.

> 12) Net transport in Fig. 8 is the same as already presented in Fig. 7. Please try to avoid duplication.

We agree that our choice of figures was not the best. Fig. 7 showed the mean transport in the upper and lower layers separately and we included it to enable comparison with the figures in Westerlund et al. (2022). Fig. 8, on the other hand, showed the monthly net transport in the whole water column but included also the southward and northward components of the net transport. To avoid duplication but still show the difference between the upper and lower layers, we decided to change these figures in the revised manuscript. Now the time series plot that was Fig. 7 is removed and the new Figs. 7 and 8 show the monthly net transports as well as the southward and northward components for the upper and lower layers, respectively. Parts of Section 5 are modified/rearranged accordingly (lines 216–239, line 262).

> 13) Figures 8 and 9 contain the same information, only for the two different transects. By such presentation, comparison of transports is not straightforward. Please consider some other reader-friendly presentation.

We agree that it was not a reader-friendly presentation. Now that we changed Figs. 7 and 8 in the revised version, we included the panel (a) from the old Fig. 8 to Fig. 9, making the comparison between the two transects easier.

> I recommend an editorial revision of the MS.

---

## Author Comment (AC2)

**Author's reply to referee comments**

Miettunen, E., Tuomi, L., Westerlund, A., Kanarik, H., and Myrberg, K.: Transport dynamics in a complex coastal archipelago, EGUsphere [preprint], https://doi.org/10.5194/egusphere-2023-1547, 2023.

Below, referee comments are displayed with italic font, highlighted with a grey background. Author replies are without highlighting. When listing the changes in the revised manuscript, line numbers refer to the "Author's tracked changes" file.

**RC2, Anonymous Referee #2, 11 Oct 2023**

> *General comments:*

> *The manuscript is very well structured, of appropriate length, and has nice high-quality figures, in my opinion. Further, the text is very easy to read and contains relatively few grammatical or spelling mistakes. So, from this point of view I'm perfectly happy with the manuscript as it is. However, I do have a couple of specific comments regarding science; see below.*

We would like to thank the referee for the encouraging feedback and for the comments on how to improve our work. Please find our replies to the specific comments below.

> *Specific comments:*

> *You state in the Model Description (lines 78-80) that open boundary data are from a CMEMS physical reanalysis; can you elaborate on this? Which variables are prescribed at the boundary, and at what time resolution?*

> *Later in the manuscript you state (lines 273-274) that daily averages are used at the open boundary. Surely, SSH data must be available on hourly resolution? How about currents? I'm a little bit concerned that the barotropic transports are affected by the use of daily averages instead of e.g. hourly data. To address this issue, I propose you validate sealevels against tide gauge data inside the computational domain, preferably one station in the south and one in the north, and perhaps validate the differences between these two (model vs. observations in both cases). If the lack of hourly data at the boundary affects the performance of the model close to the boundaries, the modelled transports may still be okay in the inner domain due to the filtering effect of the archipelago, and you may be able to show this in case you have at least one tide gauge in the inner part of the domain (e.g. Turku?). This may be enough if there are no suitable tide gauges near the northern and southern boundaries. If this is not possible, there will be some lingering doubts about the validity of the model setup with the boundary conditions being used (daily averages), and in extension, the conclusions.*

> *As an alternative, would it be possible to rerun the model for a short time period using hourly resolution of the open boundary conditions, and compare with daily averages?*

Thank you for pointing out that the open boundary data was not clearly enough described in the original manuscript. Sea surface height data at the open boundaries were indeed used at 1 h intervals, and barotropic velocities, temperature and salinity at 24 h intervals. We have clarified this in the model description in the revised manuscript (lines 100–101 in the "Author's tracked changes" file).

Modelled sea surface height was validated in our earlier paper (Westerlund et al., 2022) using tide gauge data from Föglö station which is located in the western Archipelago Sea. This is the only tide gauge station that can be regarded to be representative of the overall sea level variation in the model domain. The other two (Forsmark on the Swedish coast and Turku on the Finnish coast) represent local conditions. The validation showed that the model is able to reproduce the sea level variations quite well, indicating that barotropic dynamics are reliably reproduced.

We now realise that this earlier validation should have been referenced more clearly in this manuscript. In the revised version, we now summarise the SSH validation done in Westerlund et al. (2022) at the beginning of Section 3 (lines 140–143).

> *Technical corrections:*

> *Caption for Figure 7: Please add the info that these are northward volume transports (as I think they are).*

This figure no longer exists in the revised manuscript because it was replaced with another figure (as explained in the next reply).

> *Line 184: It seems to me that the mean transport in the lower layer in the central transect is close to zero..? (green curve, middle panel)*

The mean transport in the lower layer in the central transect is indeed close to zero, but nevertheless slightly northward (not southward, as it was written by mistake in the original manuscript). Northward net transport in the lower layer of the central transect is now more clear in the revised manuscript. To avoid duplication of figures (brought up by the Referee 1) and also to show the differences between the upper and lower layers and different transects more clearly, we have replaced Figs. 7 and 8 with new figures showing the monthly net transports and their northward and southward components separately for the upper and lower layers. The new Fig. 8b in the revised manuscript now shows that while the net transport in the lower layer in the central transect is small, it is nevertheless mostly northward.

> *Lines 253 (and other places): Change "Baltic Proper" -> "Baltic proper" (lower-case "p"). Throughout most of the manuscript you spell with lower-case "p", which is grammatically correct I think (though some authors use capital "P"). However, it is important to be consistent.*

Yes, it should be with lower-case "p". These are corrected in the revised text.

> *Line 313: Please check if this reference should perhaps be "Westerlund et al. (2021), as on github?*

We have checked this. The reference is to the GitHub repository. The earlier paper is cited elsewhere in the manuscript.

---

## Author Comment (AC3)

**Author's reply to community comments**

Miettunen, E., Tuomi, L., Westerlund, A., Kanarik, H., and Myrberg, K.: Transport dynamics in a complex coastal archipelago, EGUsphere [preprint], https://doi.org/10.5194/egusphere-2023-1547, 2023.

Below, the community comments are displayed with italic font, highlighted with a grey background. Our replies are without highlighting.

**CC1: 'Comment on egusphere-2023-1547', Maria Matos, 25 Aug 2023**

> *This review reflects comments and contributions by Júlia Sambugaro and Maria Carolina Matos resulted from the graduate-level course "How to Read and Evaluate Scientific Papers and Preprints" from the University of São Paulo, which aimed to provide students the opportunity to review scientific articles, develop critical and constructive discussions on the endless frontiers of knowledge, and understand the peer review process.*

We thank Júlia Sambugaro and Maria Carolina Matos for taking time to write community comments to our manuscript and help us improve our work.

> *The preprint examines circulation and water transport in the Archipelago Sea, Finland. Earlier works in the study area failed to estimate the water transport due to the complex topography of the archipelago, leading to an overestimation. The vulnerability of this region justify the necessity of understanding the transport dynamics.*

> *Using a high-resolution NEMO configuration, the study reveals that currents are influenced by the area's geography, resulting in stronger currents in narrow channels and weaker currents in more open spaces. Seasonal and interannual variations in transport volume and direction were observed, emphasizing the intricate dynamics of the Archipelago Sea's water exchange processes.*

> *The work rectifies past limitations and offers valuable insights for managing this unique marine ecosystem, presenting an advancement in understanding transport dynamics in the Archipelago Sea.*

> *MAJOR COMMENTS*

> *Introduction*

> *Consider adding a concise statement about the main research goal to improve the last paragraph that previews of what the preprint aims to achieve.*

Thank you for the suggestion. We aim at expressing the main research goal more precisely in the revised version.

> *Your study group has been conducting various research projects in the region, contributing to a better understanding of such a complex area. We believe that in order to reach broader audiences, it would be beneficial to include a map depicting a larger surrounding area, along with a reference to the corresponding country in the text (beginning of the Introduction). With this addition, the current Figure 1 in the introduction could be relocated to the Methods section, where it would fit more appropriately.*

It is true that not all our potential readers know the Baltic Sea region beforehand. However, we think that including an additional map would unnecessarily increase the number of figures. In the revised version, we added the names of the surrounding countries to the smaller map to make it more informative.

> *Methods*

> *There is a concern regarding observing seasonal variations through a model validated using measurements that don't have complete seasonal data coverage. Maybe in the future, collect data seasonally would bring more reliability to modeled seasonal variations.*

We agree that temporally (and spatially) more extensive observation dataset would be helpful in model development and validation. Our validation dataset includes measurements both from autumn/winter, when the water column is mixed, as well as from summer when the water column is thermally stratified. Even

though for example the timing of thermocline formation cannot be validated, comparison with the available data gives some indication that the model can reproduce both mixed and stratified conditions. Moreover, in our earlier paper (Westerlund et al., 2022), we present validation of the modelled sea surface height against continuous water level measurements, which indicates that the model is able to describe the barotropic dynamics in the model area quite well. This earlier SSH validation is now summarised at the beginning of Section 3 in the revised manuscript.

> *Model validation*

> *Another concern revolves around the differences in depth between what was measured and modeled. Despite the RMSE being lowest for the first ten meters, the magnitude of the value is high enough to raise doubts about the ability of the model to predict data.*

These are typical RMSE values for circulation models of this type and for this region. Considering the range of yearly temperature variations in this region (from freezing point to approx. 20°C), we think these values are acceptable.

> *It was mentioned that "Direction distribution of the modeled currents is slightly narrower than that of measured currents both in Norrgrundet and Utö (Fig. 2)", perhaps the bigger discrepancy between the current speed and direction should also be mentioned.*

The differences between measured and modelled currents in both stations are discussed in more detail in the paragraphs following the above-mentioned sentence.

> *Currents in the Archipelago Sea*

> *Very interesting that wind can be represented by mean values while currents can't due to its bi-directional nature.*

It is true that we refer to the long-term wind distribution as "long-term mean". However, the wind roses include all wind directions and magnitudes from the presented period, not mean values.

> *Discussion*

> *To show the interannual variation was presented the wind and its variability data. Therefore to show the seasonal variation the importance of wind and fluvial discharge was mentioned. So, it would be helpful to provide the fluvial discharge data.*

As the focus of this study is on the overall circulation dynamics of the Archipelago Sea region rather than the different components of the water budget of the Gulf of Bothnia, we think that including the river discharge data is not needed here. The river discharges to the Gulf of Bothnia have their maxima during the spring and early summer due to melting of snow in the catchments. Of course, this is not obvious to all readers from outside the Baltic Sea region, so we added a reference to support this statement in the revised manuscript.

> *It is great that you recognize the uncertainties of the model and already suggest the next improvement in the method: "As noted already by Westerlund et al. (2022), one way to address the issues caused by the boundary conditions would be to develop a two-way nested configuration with a coarse resolution Baltic Sea model and the high-resolution local model."*

Thank you. This is indeed an important development step for us to work on.

> *MINOR COMMENTS*

> *It is mentioned in the text "Fig. 1b", but in the figure itself the maps are not named "a" or "b".*

Thank you for pointing this out. We added the missing labels for subfigures in the revised version.

> *The smaller map in Figure 1 could be improved by coloring the land masses another color to make it distinct from the sea area.*

We modified the map for the revised version and marked the land with the same grey colour that is used in the bigger map.